# Prospective Validation of Indocyanine Green Lymphangiography Staging of Breast Cancer-Related Lymphedema

**DOI:** 10.3390/cancers13071540

**Published:** 2021-03-26

**Authors:** Mads Gustaf Jørgensen, Navid Mohamadpour Toyserkani, Frederik Christopher Gulmark Hansen, Jørn Bo Thomsen, Jens Ahm Sørensen

**Affiliations:** 1Department of Plastic Surgery, Research Unit for Plastic Surgery, Odense University Hospital, 5000 Odense, Denmark; fhans16@student.sdu.dk (F.C.G.H.); Joern.Bo.Thomsen@rsyd.dk (J.B.T.); jens.sorensen@rsyd.dk (J.A.S.); 2Clinical Institute, University of Southern Denmark, 5000 Odense, Denmark; 3OPEN, Open Patient data Explorative Network, Odense University Hospital, 5000 Odense, Denmark; 4Department of Plastic Surgery and Burns Treatment, Rigshospitalet, 2100 Copenhagen, Denmark; navid.toyserkani@regionh.dk

**Keywords:** lymphedema, indocyanine green, lymphangiography, breast cancer, observer

## Abstract

**Simple Summary:**

Indocyanine green lymphangiography (ICG-L) allows real-time investigation of lymphatics; however, the applicability in evaluating breast cancer-related lymphedema (BCRL) is sparse and not well established. In this prospective study, we aimed to validate ICG-L assessment of BCRL in a large patient group. We found that evaluation of BCRL with ICG-L was easy and safe to perform in the outpatient clinic and provided unique disease information unobtainable by clinical assessment alone. Future studies that evaluate the efficacy of therapeutic treatments on lymphatic function morphology should incorporate lymphatic imaging as an outcome.

**Abstract:**

Indocyanine green lymphangiography (ICG-L) allows real-time investigation of lymphatics. Plastic surgeons performing lymphatic reconstruction use the ICG-L for patient selection and stratification using the MD Anderson (MDA) and the Arm Dermal Backflow (ADB) grading systems. However, the applicability of ICG-L in evaluating breast cancer-related lymphedema (BCRL) is sparse and not well established. This study comprehensively examines the usability of ICG-L in the assessment of BCRL. We prospectively performed ICG-L in 237 BCRL patients between January 2019 and February 2020. The aim of this study was to assess the interrater and intrarater agreement and interscale consensus of ratings made using the MDA and ADB scales. Three independent raters performed a total of 2607 ICG-L assessments. The ICG-L stage for each grading system was correlated to the lymphedema volume to assess the agreement between the ICG-L stage and clinical severity. The interrater agreement was near perfect for the MDA scale (kappa 0.82–0.90) and the ADB scale (kappa 0.80–0.91). Similarly, we found a near-perfect intrarater agreement for the MDA scale (kappa 0.84–0.94) and the ADB scale (kappa 0.88–0.89). The agreement between the MDA and the ADB scales was substantial (kappa 0.65–0.68); however, the ADB scale systematically overestimated lower ICG-L stages compared to the MDA scale. The volume of lymphedema correlated slightly with MDA stage (Spearmans rho = 0.44, *p* < 0.001) and ADB stage (r_s_ = 0.35, *p* < 0.001). No serious adverse events occurred. The staging of BCRL with ICG-L is reliable, safe, and provides unique disease information unobtainable with clinical measurements alone. The MDA scale seems to provide better disease stratification compared to the ADB scale.

## 1. Introduction

Indocyanine green lymphangiography (ICG-L) is increasingly being used for evaluating lymphedema prior to microsurgical treatment of lymphedema [1,2,3]. Microsurgical lymphedema treatment, such as lymphovenous anastomosis, guided by ICG-L staging has shown varied success and results [4]. This may be attributed to poor patient selection due to subjective-, user-, and protocol-dependent ICG-L assessments. Developing an accurate, standardized, and reproducible staging system is essential for uniform interpretation of ICG-L images. Plastic surgeons performing lymphatic reconstruction conduct preoperative ICG-L to stratify and select patients for surgery using either the Arm Dermal Backflow scale (ADB) [5,6] or the MD Anderson classification (MDA) [1,2]. Currently, there is no consensus as to which scale is the more appropriate and the validity of the scales is unknown [7]. The development of the ADB scale was based on the examination of 20 patients, and the MDA was based on 30 patients [2] and later modified after evaluating another 19 patients [1]. The ADB and MDA staging systems both comprise of six stages used to grade the degree of lymphedema severity from 0–5, where 0 is normal linear lymphatics with no dermal backflow (Table 1). Stages 1–5 depict abnormal lymphatic patterns with various degrees of dermal backflow. Both the MDA and ADB staging systems are widely used for BCRL staging and evaluation of surgical outcomes [1,2,3,5,6,8,9,10,11,12]; however, neither classifications have undergone validation.

The primary aim of this study was to assess the interrater and intrarater agreement and interscale consensus between ratings using the ADB and MDA scales in a large BCRL cohort. The secondary aim was to compare the ADB and MDA scales’ applicability for stratification of BCRL patients.

## 2. Materials and Methods

### 2.1. Participants

This study is a cross-sectional study of all breast cancer-related lymphedema patients evaluated at our clinic for experimental lymphedema treatment between January 2019 and February 2020. We conducted the study according to the STROBE statement [13]. Prior to inclusion, all referred patients were screened for eligibility and invited for participation based on the following criteria:Unilateral arm lymphedema diagnosed clinically by a lymphedema physiotherapist;Unilateral arm lymphedema previously treated with completed decongestive therapy by a lymphedema physiotherapist. This treatment consisted of manual lymphatic drainage, skincare, exercise, and bandaging at the lymphedema physiotherapist’s discretion. Following complete decongestive therapy, patients were fitted with a custom-made compression sleeve intended to be worn during the daytime. Patients with lymphedema affecting the hand were additionally fitted with a compression gauntlet. Severe lymphedema cases were also fitted with a night compression sleeve and/or treated by a pneumatic compression device;History of loco-regional breast cancer treated with axillary lymph node dissection;Recurrence-free and cancer-free for more than one year;Lymphedema for more than one year;No previous surgery for lymphedema;American Society of Anesthesiologists Classification 1 or 2;Body mass index (BMI) ≤ 35;Able to communicate in oral and written Danish;No history of other malignancy apart from breast cancer and non-melanoma skin cancer;Healthy contralateral arm for comparison;No insulin-dependent diabetes;No known hepatitis, HIV, or syphilis infection;No primary lymphedema or non-breast cancer-related lymphedema;No known allergy to iodine (contraindication for ICG lymphangiography).

All included patients signed an informed consent, underwent ICG-L, arm volume estimation, clinical evaluation, and a detailed history was obtained.

In addition, we registered the following demographic information for each patient: age, relationship status, employment status, time of lymphedema diagnosis, previous arm cellulitis since lymphedema diagnosis, arm laterality of lymphedema, arm dominance, and breast cancer treatment and use of conservative lymphedema treatment (compression sleeve, gauntlet, night compression, and pneumatic compression devices). The patient’s current weight and height were measured in the outpatient clinic, and the body mass index was calculated. The following information regarding previous breast cancer treatment was retrieved from the Danish Breast Cancer Group registry [14]: type of breast surgery, radiation therapy, chemotherapy, and the number of lymph nodes removed by axillary dissection.

### 2.2. Indocyanine Green Lymphangiography

The ICG was injected subcutaneously and intradermally at the first and third webspace and on the ulnar border of the palmaris longus tendon at the level of the wrist. The ICG-L was performed after lymphedema volume assessment by the first author. We injected 0.1 mL ICG (2.5 mg/mL Verdye, Diagnostic Green, Ascheim, Germany) into each injection site. We performed three separate whole limb scans after injections using the HyperEye medical system (MNIRC-501, HEMS; Mizuho Co., Tokyo, Japan) at the following time points: 0 min, 10 min, and 1 h. The three scans were saved as video recordings with a duration of 1 min 30 s to 2 min 30 s. The first scan was recorded immediately after injection. Then, the patients were instructed to lie still for 10 min, and the second scan was recorded, including dynamic ICG velocity assessment using the method described by Yamamoto et al. [15]. In brief, we measured the wrist’s distance to the most proximal visible ICG pattern on the arm after 10 min. Then, the traveled distance was divided by the total arm length to account for inter-individual variations in arm lengths. After approximately 60 min, the patients underwent the third ICG scan, which was used for staging by the ADB and MDA grading systems.

Following ICG injection, we monitored all patients for at least 60 min for allergic and hypersensitive reactions caused by the dye.

### 2.3. Lymphedema Assessment

The arm volumes were measured on both the lymphedematous and the healthy arm using tape measurements (SECA 201, Hamburg, Germany) and deducted from each other to calculate ΔVolume. The tape measurements were performed by the first author (MGJ) prior to the ICG-L assessment, clinical exam, and patient history. The arm circumference was measured at the wrist, middle of the forearm, elbow, middle of the upper arm, and proximally on the upper arm. The distance of each measured segment from the wrist was measured and total arm volume calculated using the truncated cone formula [16]. The ΔVolume was defined as the volume of the affected arm minus the volume of the healthy arm.

We used the internationally accepted International Society of Lymphology (ISL) staging system to grade the BCRL severity clinically [17]. The ISL staging system was as follows, Stage 0 patients: no clinical swelling. Stage 1: slight clinical swelling that subsides with limb elevation. Stage 2: moderate clinical swelling that does not subside with limb elevation. Stage 3: elephantiasis with trophic skin changes.

### 2.4. Validation Criteria and Standards

Three independent raters reviewed the ICG-L scans recorded after 60 min (rater 1: MGJ (very experienced), rater 2: NMT (moderate experience), rater 3: FCGH (limited experience). The scans were semi-quantitatively graded visually using the ADB and MDA staging systems by all raters.

The interrater assessments were performed by comparing the assessment of rater 1, rater 2, and rater 3. Rater 2 and rater 3 performed the ratings while blinded for the patient’s clinical exam and history. The interrater assessments were used to establish a frequency baseline for grading’s within each staging system and whether grading’s differed between raters and the ICG-L experience of raters.

Intrarater assessments were performed by rater 2 and rater 3, who reviewed the third scans a second time after a two-month wash-out period. Rater 2 and rater 3 also performed these ratings while being blinded to the results of the patient’s clinical exams and history as well as their previous assessments. This intrarater assessment established whether gradings differed within raters and the reproducibility of the assessments.

Interscale agreements were performed by comparing the ADB and MDA stages as graded by rater 1, rater 2, and rater 3. This assessment established a consensus between the staging systems and determined whether patient stratification varied according to the type of grading system used.

The assessments were standardized a priori due to all raters grading the same ICG-L scan recordings. As multiple ADB and MDA gradings can exist within one arm, we used the highest observable stage within the arm as the final grade for both scales. The presence of dermal backflow patterns was defined as any non-linear abnormal dermal backflow pattern in the limb, irrespective of its size and fluorescence intensity. The three independent raters performed a total of 2607 assessments including grading of both the MDA and ADB scale in inter- and interrater assessments.

### 2.5. Statistical Methods

We described the baseline characteristics with means ± standard deviation (SD) for continuous parametric variables, median and interquartile range (IQR) for nonparametric variables, and rounded frequencies (%) for categorical variables. The skewness/kurtosis test was used to test for normal distributions of continuous variables. We used Cohen’s kappa statistics to compare and validate the MDA and ADB inter- and intrarater agreement assessments. Cohen’s kappa was also used to compare the overlap of MDA and ADB scales by performing an interscale agreement between scales. We a priori categorized the following thresholds for kappa agreements [18]:

0.00–0.10 = poor agreement;

0.11–0.20 = slight agreement;

0.21–0.40 = fair agreement;

0.41–0.60 = moderate agreement;

061–0.80 = substantial agreement; 

0.81–0.99 = near perfect agreement;

1.00 = perfect agreement.

The correlation between the MDA and ADB stages and the ISL stage, lymphedema volume, and ICG velocity was conducted using Spearman’s rho correlation coefficient (r_s_) with the following a priori defined thresholds [19]:

0.00–0.30 = poor correlation;

0.31–0.50 = slight correlation;

0.51–0.70 = moderate correlation;

0.71–0.90 = substantial correlation;

0.91–0.99 = near-perfect correlation;

1.00 = perfect correlation.

We categorized the ICG grades binary for both the MDA and ADB scales as the early ICG stage (ICG stage 0–2) and advanced ICG stage (ICG stage 3–5).

To compare the MDA and ADB scale’s applicability for stratification of patients with lymphedema, we compared clinical variables and lymphedema volumes between early and advanced ICG stages. Comparisons between early and advanced stages for both MDA and ADB classifications were performed using unpaired *t*-test, Chi-squared, or Mann–Whitney test depending on data type and distribution.

STATA 15 (StataCorp. 2017. Stata Statistical Software: Release 15. College Station, TX, USA) was used for the statistical analysis and conducted with a two-tailed significance level of 0.05 and reported with 95% CI when applicable.

## 3. Results

We included 237 BCRL patients in this study (Figure 1 and Table 2). Three observers performed interrater assessments, and two observers performed intrarater assessments using both the MDA and ADB scales for a total of 2607 assessments (Table 3). The interrater agreement was near perfect for the MDA scale (kappa 0.82–0.90) as well as the ADB scale (kappa 0.80–0.91) with no systematic bias (Figure 2). Similarly, we found near-perfect intrarater agreement for the MDA (kappa 0.84–0.94) and the ADB scale (kappa 0.88–0.89) with no systematic bias (Figure 3). The interscale agreement of the MDA and ADB scales was substantial (kappa 0.65–0.68), however, the Bland–Altman plots revealed that the ADB systematically overestimated the early stages of lymphedema (Figure 4).

Eleven patients (4.64%) had stage 0 with no dermal backflow when assessed by both the MDA and ADB grading systems (Figure 5, Appendix A). We found 14 patients with MDA stage 1 with minimal dermal backflow (Appendix A); however, we did not find any patients with ADB stage 1 with dermal backflow around the axilla only. More patients were graded as stage 2 on the MDA scale with segmental dermal backflow (42 patients (17.72%), Appendix A) compared to stage 2 on the ADM scale with dermal backflow in the upper arm only (14 patients (5.91%), Appendix A). Fewer patients were graded as stage 3 on the MDA scale with a substantial backflow (79 patients (33.33%), Appendix A) compared to stage 3 on the ADB scale with dermal backflow involving the forearm (123 patients (51.90%), Appendix A). The number of patients who had dermal backflow involving the hand was equal in both scales (85 patients (35.86%), Appendix A). Six patients (2.53%) were categorized as MDA stage 5, with no visible ICG proximal to the injection site (Appendix A). However, we did not find any stage 5 ADB patients with the diffuse pattern involving the entire limb. Six patients (2.53%) did not conform to the ADB staging system as no ICG flow was detected proximal to the injection site.

More patients staged by the MDA scale were in the early stages 0–2 compared to the ADB scale (28.27% vs. 9.96%, *p* < 0.001, Figure 5A,B). Patients with advanced MDA ICG stage 3–5 were slightly older (61.01 ± 9.23 years vs. 56.26 ± 9.92 years, *p* < 0.001), and slightly more patients were unemployed (54.12% vs. 45.88%, *p* < 0.05), had a longer duration of lymphedema in years (4.86 (5.76) vs. 3.67 (5.01), *p* < 0.05), and more patients had a history of cellulitis (41.18% vs. 17.91%) compared to patients with MDA ICG stage 0–2 (Table 4). Patients with advanced ADB ICG stage 3–5 had longer latency until lymphedema onset in years (0.47 (0.59) vs. 1.92 (1.55), *p* < 0.05), and more patients had a history of cellulitis (37.02% vs. 8.70%) compared to patients with ADB ICG stage 0–2.

There was a poor correlation between ISL stage and MDA (r_s_ = 0.27, *p* < 0.05) and ADB (r_s_ = 0.17, *p* < 0.05) stages, suggesting a high disagreement between clinical and lymphographical staging. Increased lymphedema volume was only slightly correlated to an advanced ICG stage in both the MDA (r_s_ = 0.44, *p* < 0.001) and ADB (r_s_ = 0.35, *p* < 0.001) assessments (Figure 5C,D). The dynamic ICG velocity was poorly correlated with the MDA stage (r_s_ = −0.19, *p* < 0.05) and ADB stage (r_s_ = −0.25, *p* < 0.05, Figure 5E,F), suggesting that the velocity of ICG does not affect the dermal backflow patterns.

Three patients (1.27%) were administered over-the-counter antihistamine due to uncomfortable itching following ICG injections. No patients developed shortness of breath, tachycardia, or hypotension, and no patients returned with cellulitis following the ICG injections.

## 4. Discussion

In this study, we found near-perfect inter- and intrarater agreement when assessing the applicability and reliability of the MDA and ADB scales for ICG-L staging of 237 BCRL patients. There was a substantial agreement between the MDA and ADB scales in the assessment of interscale reliability when compared using kappa statistics; however, the ADB overestimated the ICG-L stage in milder lymphedema cases. The volume of lymphedema showed a moderate correlation to the ICG stages. ICG staging was safe to perform in the outpatient clinic, well-tolerated, and feasible within one hour.

Currently, there is no standard regarding injection technique and time to be used for ICG-L staging in the literature. In this study, our injection technique varied slightly from the original MDA and ADB studies. In the original ADB study, the ICG depots were injected at the wrist near the ulnar border of the palmaris longus tendon and into the second web space of the hand. In the original MDA study, the authors injected ICG into all finger web spaces and made no injections near the wrist. We chose to fuse the two injection techniques to accommodate both methods for comparison in this study, we injected in the first and third webspace and one injection at the wrist near the palmaris longus tendon’s ulnar border.

There were some dissimilarities when comparing the staging results using the MDA and the ADB scales. The majority of patients had a variable extent of dermal backflow involving the forearm, and as such, more than 90% of patients were classified as ADB stage 3 or higher. Thus, the most considerable discrepancy between the two scales was that the MDA stage 1 and 2 patients with minimal or segmental backflow in the forearm were classified as ADB stage 3 due to backflow located in the forearm. We did not find any patients having ADB stage 1, dermal backflow around the axilla only, and we did not find any patients with ADB stage 5, diffuse pattern involving the entire limb. This could be due to differences in lymphedema duration as well as time used for ICG-L assessment, which was not described in the original ADB study. All of our patients had stabilized lymphedema for more than one year. It can thus be speculated if ADB stage 1 only occurs in patients with newly developed lymphedema. The time used for ICG-L assessments was one hour in this study, and it is possible that a shorter time for assessments may alter the ICG-L stage due to the limited time for migration of the dye. Interestingly, we observed no migration of the ICG dye in 6/237 patients (2.53%). This incidence compares to the findings of Akita et al. in a BCRL screening study (5/205 patients (2.44%)) [20]. Akita et al. performed re-injections in the patients with no flow and found migration of the dye in the second and the authors concluded that the no proximal flow is a temporary observed state. We were unable to confirm the findings by Akita et al., as we did not perform re-injections in patients with no ICG-L flow. During this study, we observed that patients had a very variable extent of their dermal backflow with each ICG-L stage as assessed by the MDA and ADB grading systems. This was especially pronounced in stage 4 patients, which had a variable amount of dermal backflow involvement in the forearm and overarm. Some patients had extensive dermal backflow in the entire arm, while others had very limited or no proximal dermal backflow in the forearm or overarm. Based on this observation, we hypothesize that the MDA and ADB stage 4 classification is too broad and that these patients may benefit from additional stratification based on the proximal backflow patterns. Future studies may be needed to provide additional stratification of these patients

Our study found several indicators for beneficial early microsurgical lymphedema treatment to prevent disease progression [21]. For example, there was a trend that patients with shorter time until lymphedema diagnosis had normal lymphatics or only mild dermal backflow located in the upper arm when assessed by ICG. In contrast, patients with longer lymphedema duration, cellulitis, and increased age seemed to have more extensive dermal backflow. These findings suggest that the risk of disease progression is imminent over time despite conservative management and highlight the importance of surgical prevention and early treatments [22,23,24]. Early identification of patent lymphatics seems crucial for successful lymphovenous anastomoses when treating lymphedema [2,25]. Arm cellulitis, increased patient age, and lymphedema duration may, therefore, increase the likelihood of advanced ICG-L dermal backflow and associated subdermal lymphatic vessel sclerosis, which can hamper the feasibility and efficiency of microsurgical lymphedema treatments [26,27].

This study’s strength is its large patient size and the number of blinded observer assessments, with a comprehensive correlation to patient demographics. In our study, we found that lymphedema volume only slightly correlated with ICG-L stage. This finding is confirmed by two American studies of patients with extremity lymphedema of mixed etiology [9,11] and a recent study from Belgium including 45 BCRL patients [28]. All patients in this study had received lymphedema treatment by a physiotherapist prior to inclusion in the study and received yearly renewals of fitted compression garments. BCRL is a disease that demands a multidisciplinary approach and treatment including complete decongestive therapy and compression garments are often fitted and adjusted shortly after BCRL diagnosis, to reduce and maintain the volume of BCRL [29]. Physiotherapeutic lymphedema treatment is currently the first line and cornerstone of lymphedema treatment and at our institution patients are not evaluated for surgical intervention before an optimized conservative treatment has been established. However, surgical management of lymphedema is moving towards the prevention of lymphedema and this paradigm may be changing [21]. Due to the current conservative treatment regimen, these findings can be generalized to all conservative treated BCRL patients seen for surgical evaluation. For this reason, we decided not to include patients with a BCRL duration of less than one year in this study. This was also the reason that we had no baseline measurements of the affected or unaffected arm prior to BCRL onset or breast cancer treatment. Assessing all patients at the time of breast cancer treatment would also be ethically ambiguous because we would put an additional burden on the patient and end up measuring many patients that would never go on to develop BCRL. Regardless, lymphatic imaging studies have shown that BCRL patients have systemically reduced lymph drainage from the unaffected arm and lower extremity [30,31]. This raises the possibility that the local lymphatic injury can have systemic manifestations or that BCRL may be a predisposed condition [32,33]. The predisposition of BCRL is interesting and the Mascagni-Sappey lymphatic pathway and available presence of lymphaticovenous communications may be a protective factor against BCRL [34,35]. In this study, we did not investigate the Mascagni-Sappey pathway, as it would involve ICG injections near the cephalic vein and this may have conflicted with the ADB and MDA grading systems. A limitation of this study is that we graded the ICG-L recordings at fixed time points. Therefore, we could not assess lymphatic vessel functionality such as the frequency of lymphatic vessel pumping or propulsion. These may be important parameters for lymphatic morphology, however, the usability of this has yet to be inspected in lymphedema patients [36]. So far, findings in healthy individuals have shown that lymphatic pumping frequency and propulsion are unreliable to quantify [37]. This limitation may also be somewhat insignificant, as the purpose of the study was to assess the validity of ICG-L in assessing BCRL at a single point in time. Missing baseline arm measurements prior to breast cancer treatment is a limitation inherent to all BCRL literature and this study was no exception. This limitation of missing baseline measurements at the time of breast cancer treatment is however unlikely to improve in the future due to the impracticality of preemptive lymphedema measurements at the time of breast cancer diagnosis. Also, patient arm volumes are subject to change over time dependent on the patient’s physical activity level, age, and BMI. While it would be commended to perform arm measurements at the time of breast cancer diagnosis, it would lead to over-measuring 2/3 of patients that never will develop BCRL. In clinical reality, very few patients undergo clinical or ICG-L measurements at the time of breast cancer treatment and lymphatic surgeons are most often approached by patients that have no prior measurements available. Therefore, these study methods and results are highly relevant to clinical reality.

BCRL is almost synonymous with limb swelling and several methods exist to quantify lymphedema volume. Water-displacement is a precise method for measuring the volume of BCRL in a research setting. However, the disadvantages of water-displacement are that they are costly and cumbersome to use in daily practice [16]. We have water-displacement facilities to measure BCRL volume at our department [29], however, few other centers have access to similar facilities and this makes direct translations and clinical interpretations across centers difficult [38]. Perometry is another technical method for measuring BCRL; however, perometry measurements have shown inconsistent accuracy depending on the arm position and it has translational disadvantages to centers without perometry. Similarly, three-dimensional laser scanning can quantify arm volumes, however, this method is also dependent on arm positioning similar to perometry. Circumferential tape measurements on the other hand are universally available and are the most commonly used measurement for BCRL with a high correlation to water-displacement, perometry, and three-dimensional laser measurements [9,16,39]. The main limitation of tape measurements is the inconsistency in tightening the tape measure around the arm. This limitation was circumvented in this study, as we used a flexible no-stretch spring-loaded tape measure to ensure equal tightness on the skin for all measurements. We found a poor to slight correlation between ISL stage, lymphedema volume, and the ICG-L. This is in complete agreement with Chang et al., who also found a poor correlation between the ISL stages and ICG-L stages [11]. This evidence suggests that ICG-L provides unique disease information, not conceivable with clinical measurements alone. Given the agreement between tape measurements, water-displacement, and perometry, it is unlikely that another method for volume assessment would yield a stronger correlation [38]. We use the ICG-L stage to guide lymphovenous bypass surgery using similar criteria as other large published cohort studies [3,40]. Patients with stage 0 presenting with only linear lymphatics without dermal backflow are not offered lymphovenous bypass, as there is no evident pathological lymph flow to bypass. We consider stage 1–2 patients to be prime candidates for lymphovenous bypass because they have easily definable pathological dermal backflow patterns with distal lymphatics. We have not yet performed lymphovenous bypass in patients with MDA stage 3–4 because they historically have been associated with fewer identifiable lymphatics and worse treatment outcomes compared to stages 1–2 [40,41]. Applying this treatment algorithm for the MDA stage results in 30% of our patients being eligible for lymphovenous bypass. In contrast, applying this treatment algorithm to the ADB stage results in only 10% of patients being eligible for lymphovenous bypass. The 30% patient eligibility found in our study, is comparable to a 38% eligibility found in a first-year review from a leading American lymphatic surgery center including both microsurgical and debulking procedures [42]. Limited data are available on whether this is the right approach to surgically select patients and we aim to closely monitor our patients for at least one year after treatment. We suggest that patients with MDA stage 5 presenting with no proximal lymph flow be rescheduled for a second look after a few months washout period. Additionally, the initial ICG-L stage is per se an outcome predictor following microsurgical lymphedema treatment [40], with lower ICG-L stages being associated with more favorable outcomes compared to higher stages. A novel staging system for lymphedema is available using lymphoscintigraphy [43]. However, we believe that ICG-L has several advantages compared to lymphoscintigraphy due to its bedside usability, non-radioactive contrast agent, high resolution enabling real-time identification of static, and dynamic subdermal lymphatic vessels [25]. Nevertheless, the main limitation of ICG-L, as compared to lymphoscintigraphy, is that it can only identify dermal and subdermal lymphatics down to an approximate depth of 2 cm in a two-dimensional plane. Though, with the recent technological development of three-dimensional photoacoustic ICG-L [44], we may identify deeper lymphatics for lymphovenous bypass enabling improved image-guided protocols for lymphedema staging and treatment in the future.

## 5. Conclusions

The staging of BCRL with ICG-L is easy and safe to perform in the outpatient clinic and provides unique information about lymphedema unobtainable by clinical assessment. We compared the MDA and ADB staging systems and found both of these easy to use, with substantial overlap and near-perfect inter- and intra-rater agreements. However, in an outpatient setting with time limitations, the MDA scale seems to provide better disease stratification compared to the ADB scale.

## Figures and Tables

**Figure 1 cancers-13-01540-f001:**
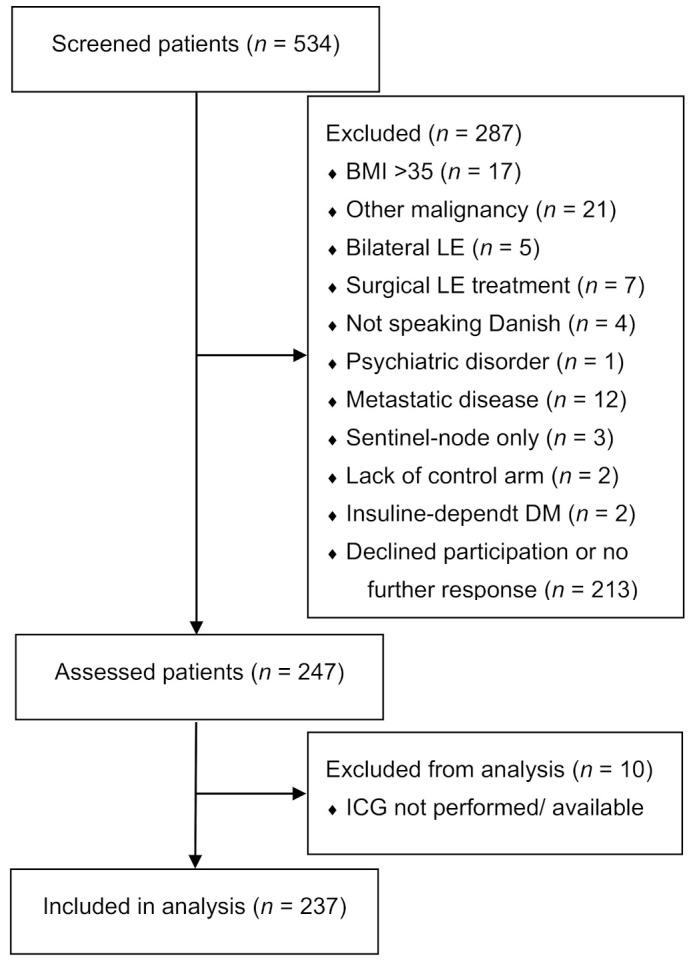
This figure shows the flowchart of included patients.

**Figure 2 cancers-13-01540-f002:**
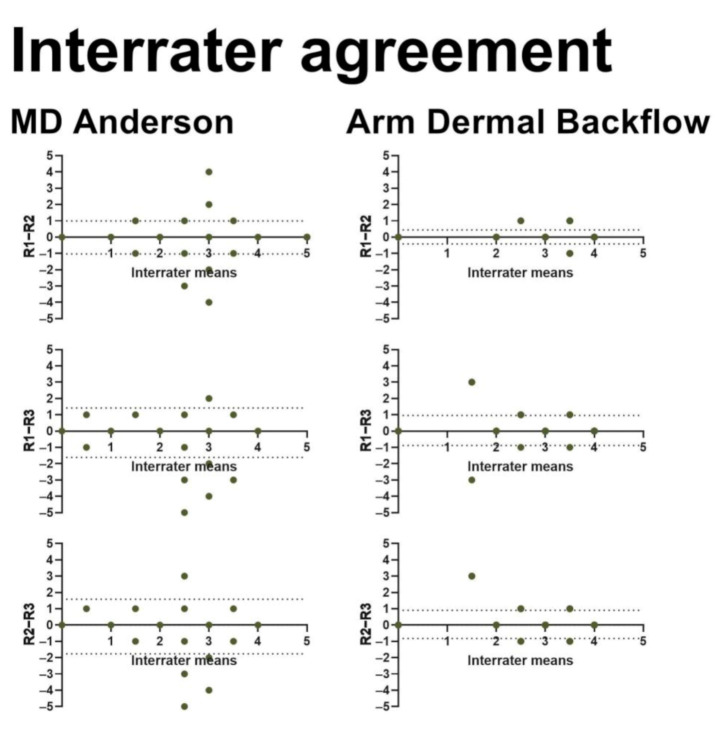
This figure shows Bland–Altman plots for interrater agreements in MDA and ADB scales. The x-axis represents the mean interrater stage (range: 0–5). The y-axis shows the difference in interrater staging (range: −5–5). R1= rater 1, R2 = rater 2, R3 = rater 3. The horizontal dotted lines denote the 95% confidence intervals for the limits of agreement.

**Figure 3 cancers-13-01540-f003:**
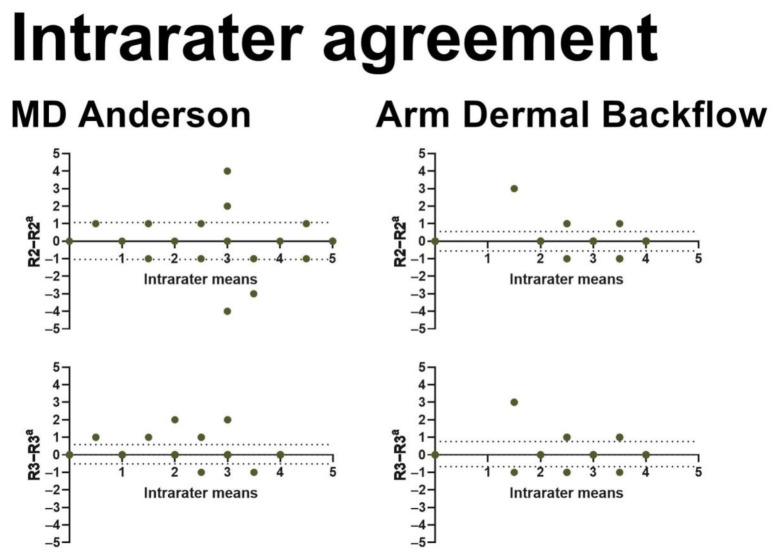
This figure shows Bland–Altman plots for intrarater agreements in MDA and ADB scales. The x-axis represents the mean intrarater stage (range: 0–5). The y-axis shows the difference in intrarater staging (range: −5–5). The horizontal dotted lines denote the 95% confidence intervals for the limits of agreement. R2 = rater 2, R3 = rater 3. a = second assessment.

**Figure 4 cancers-13-01540-f004:**
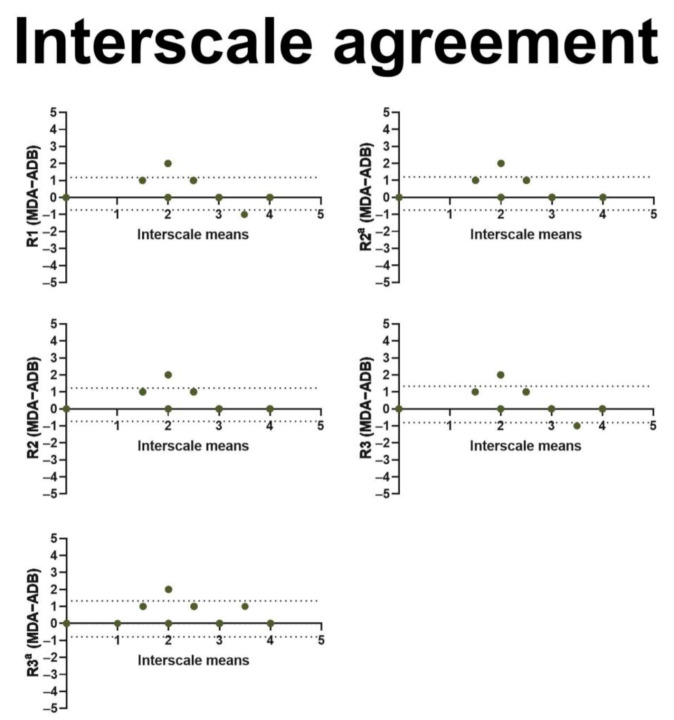
This figure shows Bland–Altman plots for interscale agreements between the MD Anderson and the Arm Dermal Backflow stages. The x-axis represents the mean interscale stage (range: 0–5). The y-axis shows the difference in interscale staging (range: −5–5). The horizontal dotted lines denote the 95% confidence intervals for the limits of agreement. R1 = rater 1, R2 = rater 2, R3 = rater 3. a = second assessment.

**Figure 5 cancers-13-01540-f005:**
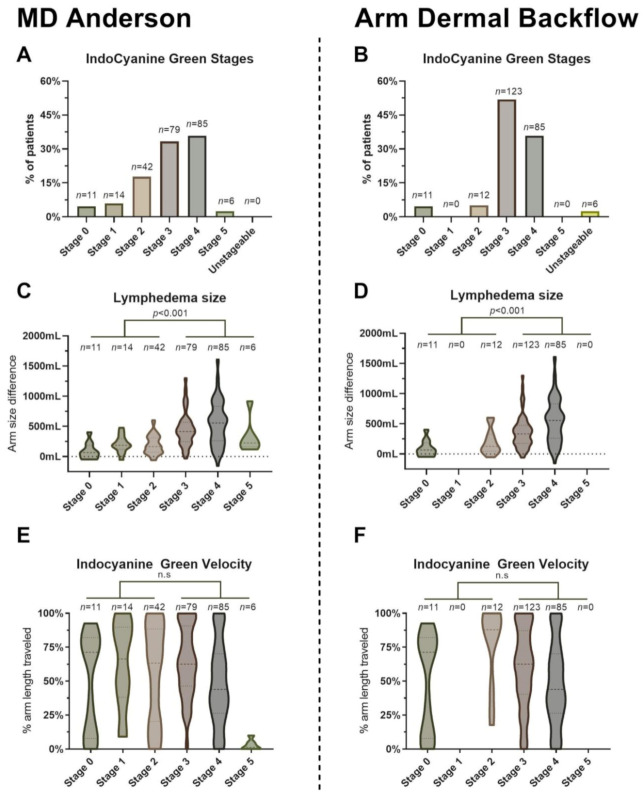
This figure shows the correlation between the MDA and ADB stages and clinical examination. (**A**) This figure shows the distribution of patients by MD Anderson stage. (**B**) This figure shows the distribution of patients by Arm Dermal Backflow stage. (**C**) This figure shows a violin plot of lymphedema volume stratified by MD Anderson stage. The thick dashed line denotes the median, and thin dashed lines denote the interquartile range. Plot thickness denotes the probability density of volumes at different values. (**D**) This figure shows a violin plot of lymphedema volume stratified by Arm Dermal Backflow stages. (**E**) This figure shows a violin plot of ICG velocity stratified by MD Anderson stage. (**F**) This figure shows a violin plot of ICG velocity stratified by the Arm Dermal Backflow stage. *n* = number of patients, *p* = *p*-value, n.s = not significant.

**Table 1 cancers-13-01540-t001:** This table shows an overview of the MD Anderson scale and the Arm Dermal Backflow scales.

Stages	MD Anderson Scale	Arm Dermal Backflow Scale
	Findings	
Stage 0	No dermal backflow	No dermal backflow
Stage 1	Many patent lymphatics and minimal dermal backflow	Splash pattern around the axilla
Stage 2	Moderate number of patent lymphatics and segmental dermal backflow	Stardust limited between olecranon and axilla
Stage 3	Few patent lymphatics with extensive dermal backflow	Stardust distal to olecranon
Stage 4	Dermal backflow involving the hand	Stardust involving the hand
Stage 5	ICG does not move proximally to injection site	Diffuse and stardust pattern involving the entire limb

**Table 2 cancers-13-01540-t002:** This table shows the patient demographics of the included patients. SD = standard deviation, N = number. IQR = interquartile range.

Variables	Data Distribution	All Patients (*n* = 237)
Age (years)	Mean ± SD	59.68 ± 9.94
In relationship (yes)	N (%)	171 (27.85%)
Employed (yes)	N (%)	140 (59.07%)
BMI (kg/m^2^)	Median (IQR)	27.44 (7.27)
Breast cancer treatment		
Radiation therapy (yes)	N (%)	223 (94.09%)
Chemotherapy (yes)	N (%)	199 (83.96%)
Endocrine therapy (yes)	N (%)	191 (80.59%)
Mastectomy (yes)	N (%)	122 (51.69%)
Post-mastectomy reconstruction (yes)	N (%)	57 (46.72%)
Abdominal free flap (yes)	N (%)	24 (42.11%)
Pedicled back flap (yes)	N (%)	17 (29.82%)
Implant-based reconstruction (yes)	N (%)	16 (28.07%)
Lymph nodes removed (No.)	Median (IQR)	17 (8)
Lymphedema characteristics		
Lymphedema latency (years)	Median (IQR)	0.71(1.42)
Lymphedema duration (years)	Median (IQR)	4.47(5.50)
Lymphedema duration: <2 years	N (%)	45 (19.99%)
Lymphedema duration: 2–3 years	N (%)	31 (13.08%)
Lymphedema duration: 3–4 years	N (%)	28 (11.81%)
Lymphedema duration: 4–5 years	N (%)	26 (10.97%)
Lymphedema duration: 5–6 years	N (%)	12 (5.06%)
Lymphedema duration: 6–7 years	N (%)	15 (6.33%)
Lymphedema duration: 7–8 years	N (%)	18 (7.59%)
Lymphedema duration: 8–9 years	N (%)	14 (5.91%)
Lymphedema duration: 9–10 years	N (%)	39 (16.46%)
Lymphedema duration: >10 years	N (%)	9 (3.80%)
Lymphedema volume (mL)	Mean ± SD	410.51 ± 326.73
Lymphedema volume (%)	Mean ± SD	18.77 ± 14.06
Lymphedema in dominant arm (yes)	N (%)	114 (48.10%)
Previous episode of cellulitis (yes)	N (%)	82 (34.60%)
Current lymphedema treatment		
Compression sleeve (yes)	N (%)	207 (87.34%)
Compression gauntlet (yes)	N (%)	133 (56.12%)
Night compression (yes)	N (%)	72 (30.38%)
Pneumatic compression device (yes)	N (%)	44 (18.57%)

**Table 3 cancers-13-01540-t003:** This table shows the interrater, intrarater and interscale agreement of the MDA and ADB scale assessments. The interrater agreement shows the agreement between raters. Intrarater agreement shows the agreement within raters. The interscale agreement shows the agreement between the MDA and the ADB scale. Interrater R2 and R3 performed staging blinded to all patient demographics and clinical variables. The second intrarater assessment was performed after a two-month wash-out interval. R1 = rater 1, R2 = rater 2, R3 = rater 3. ^a^ = second assessment.

Assessment	Agreement (%)	Expected Agreement (%)	Kappa Value	Standard Error
MDA				
Interrater agreement				
R1–R2	93.25	27.92	0.90	0.04
R1–R3	91.98	28.29	0.88	0.04
R2–R3	87.76	28.10	0.82	0.04
Intrarater agreement				
R2–R2 ^a^	88.61	28.15	0.84	0.04
R3–R3 ^a^	95.78	28.94	0.94	0.04
ADB				
Interrater agreement				
R1–R2	94.51	40.01	0.91	0.05
R1–R3	89.45	41.31	0.82	0.05
R2–R3	88.61	41.65	0.80	0.05
Intrarater agreement				
R2–R2 ^a^	93.25	40.80	0.89	0.04
R3–R3 ^a^	93.25	43.70	0.88	0.05
Interscale agreement				
R1	77.22	31.54	0.66	0.04
R2	75.95	31.28	0.65	0.04
R3	77.22	32.56	0.66	0.04
R2 ^a^	78.48	32.93	0.68	0.04
R3 ^a^	77.64	33.70	0.66	0.04

**Table 4 cancers-13-01540-t004:** This table shows the patient characteristics stratified by the MDA and ADB scales. *n* = number of patients, No. = number, ^a^ = Stage 0–2 vs. 3–5, n.s = not significant.

MDA Scale								
Variables	Data Distribution	Stage 0 (*n* = 11)	Stage 1 (*n* = 14)	Stage 2 (*n* = 42)	Stage 3 (*n* = 79)	Stage 4 (*n* = 85)	Stage 5 (*n* = 6)	Comparison *p*-Value ^a^
Age (years)	Mean ± SD	55.55 ± 9.01	58.50 ± 8.02	55.71 ± 9.74	61.85 ± 8.69	60.47 ± 10.65	57.83 ± 14.70	<0.001
BMI (kg/m^2^)	Median (IQR)	26.40 (6.59)	32.87 (8.81)	26.62 (9.03)	26.57 (6.11)	28.81 (6.78)	24.18 (17.98)	n.s
Employed (yes)	N (%)	7 (63.64%)	9 (64.29%)	32 (76.19%)	39 (49.37%)	51 (60.00%)	2 (33.33%)	<0.05
Lymph nodes removed (No.)	Median (IQR)	15 (6)	17 (9)	17 (6)	17 (8)	18 (8)	13 (7)	n.s
Lymphedema latency (years)	Median (IQR)	0.44 (1.00)	0.43 (0.76)	0.73 (0.83)	0.75 (1.78)	0.73 (1.45)	0.29 (0.55)	n.s
Lymphedema duration (years)	Median (IQR)	3.42 (6.08)	3.71 (2.25)	3.68 (5.95)	4.84 (5.39)	4.47 (6.48)	7.21 (5.61)	<0.05
Dominant arm affected (yes)	N (%)	4 (36.36%)	10 (71.43%)	18 (42.86%)	41 (51.90%)	37 (43.53%)	4 (66.67%)	n.s
Cellulitis (yes)	N (%)	1 (9.09%)	1 (7.14/)	10 (23.81%)	37 (46.84%)	30 (35.29%)	3 (50.00%)	<0.001
**ADB Scale**								
**Variables**	**Data** **Distribution**	**Stage 0** **(*n* = 11)**	**Stage 1** **(*n* = 0)**	**Stage 2** **(*n* = 12)**	**Stage 3** **(*n* = 123)**	**Stage 4** **(*n* = 85)**	**Stage 5** **(*n* = 0)**	**Comparison** ***p*** **-Value ^a^**
Age (years)	Mean ± SD	55.55 ± 9.01	N/A	61.42 ± 10.55	59.41 ± 9.23	60.47 ± 10.65	N/A	n.s
BMI (kg/m^2^)	Median (IQR)	26.40 (6.59)	N/A	33.96 (8.63)	26.59 (6.86)	28.81 (6.78)	N/A	n.s
Employed (yes)	N (%)	7 (63.64%)	N/A	5 (41.67%)	75 (60.98%)	51 (60.00%)	N/A	n.s
Lymph nodes removed (No.)	Median (IQR)	15 (6)	N/A	17(5)	17(8)	18(8)	N/A	n.s
Lymphedema latency (years)	Median (IQR)	0.44 (1.00)	N/A	0.37 (0.52)	0.79 (1.60)	0.73 (1.45)	N/A	<0.05
Lymphedema duration (years)	Median (IQR)	3.42 (6.08)	N/A	4.00 (6.38)	4.55 (5.12)	4.47 (6.48)	N/A	n.s
Dominant arm affected (yes)	N (%)	4 (36.36%)	N/A	7 (58.33%)	62 (50.41%)	37 43.53%)	N/A	n.s
Cellulitis (yes)	N (%)	1 (9.09%)	N/A	1 (8.33%)	47 (38.21%)	30 (35.295)	N/A	<0.05

## Data Availability

The data presented in this study are available on request from the corresponding author upon reasonable request.

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
