# Peer review of "Prospective Validation of Indocyanine Green Lymphangiography Staging of Breast Cancer-Related Lymphedema"

_cancers, 2021, doi:10.3390/cancers13071540_

Round 1
Reviewer 1 Report
Review 1126771
"Prospective Validation of IndoCyanine Green Lymphangiography Staging of Breast-Cancer Related Lymphedema."
This study is designed to validate two staging systems for lymphedema severity of 237 unilateral breast cancer-related lymphedema based on ICG lymphography imaging, the MD Anderson (MDA) Scale Arm Dermal Backflow (ADB) scale. They found near-perfect inter-rater and intra-rater agreement for both ICG lymphography staging and substantial agreement between the MDA and the ADB scales. Indeed, they found a slight correlation between the two ICG lymphography staging systems' results to conventional circumferential measurements. They concluded that the two ICG lymphography staging were reliable, safe tools with the MDA scale providing better disease stratification than the ADB scale.
The efforts to validate the two ICG lymphography severity staging systems to establish objective criteria to stratify BCRL patients are much appreciated, but some limitations are present in this study. Several points need to be pointed out as follows.
First, the authors stated the study's purpose is to validate two ICG lymphography staging (MDA scale and ADB scale), what were the validation criteria and standard. It is unclear if they accomplished this purpose without a more precise definition. Regarding the inter-and intra-rater agreement, the authors should mention the level of experiences of the three raters assessing the ICG images. It is not clear who evaluates (“Two observers”) the photos a second time after two months. The intra-rater evaluation should be the same assessors as the first evaluation.
Second, the authors should mention their definition of extremity lymphedema instead of generically say in the inclusion criteria, “Unilateral arm lymphedema diagnosed by a lymphedema physiotherapist”. It is not clear which protocol of CDT the patients followed before the ICG lymphography (Compression sleeve, compression garment, night compression, pneumatic compression?) and if all patients evaluated were compliant with the CDT protocol.
Third, please add information about when the tape measurements were taken and by who. The authors should clarify which statistical analysis they used for the correlation between the two ICG lymphography staging and the lymphedema size. It is interesting to note in Figure 5C that in the MDA scale, the correlation with the lymphedema size is consistent from stage 0 to stage 4 and then drop in stage 5; the authors should discuss this point.
This study is limited as measuring the lymphedema by a circumferential tape measure can significantly variability amongst measurements. A more consistent reliable mechanism to measure upper extremity lymphedema is utilizing the upper extremity's total volume via water replacement or CT volume.
Fourth, the authors should limit their assertions: “ICG-L is a superior tool compared to lymphoscintigraphy”, as there was no lymphoscintigraphy data compared.
Lastly, the authors state that the two ICG lymphography staging provides unique disease information unobtainable by clinical assessment, with an MDA scale providing better disease stratification than the ADB scale. However, this statement was not justified by the data reported. The authors should provide data on how they used the information from ICG lymphography and the disease stratification from the two ICG lymphography staging in terms of patient selection for microsurgical treatment and outcome.
Author Response
Reviewer 1:
This study is designed to validate two staging systems for lymphedema severity of 237 unilateral breast cancer-related lymphedema based on ICG lymphography imaging, the MD Anderson (MDA) Scale Arm Dermal Backflow (ADB) scale. They found near-perfect inter-rater and intra-rater agreement for both ICG lymphography staging and substantial agreement between the MDA and the ADB scales. Indeed, they found a slight correlation between the two ICG lymphography staging systems' results to conventional circumferential measurements. They concluded that the two ICG lymphography staging were reliable, safe tools with the MDA scale providing better disease stratification than the ADB scale.
The efforts to validate the two ICG lymphography severity staging systems to establish objective criteria to stratify BCRL patients are much appreciated, but some limitations are present in this study. Several points need to be pointed out as follows.
First, the authors stated the study's purpose is to validate two ICG lymphography staging (MDA scale and ADB scale), what were the validation criteria and standard. It is unclear if they accomplished this purpose without a more precise definition. Regarding the inter-and intra-rater agreement, the authors should mention the level of experiences of the three raters assessing the ICG images. It is not clear who evaluates (“Two observers”) the photos a second time after two months. The intra-rater evaluation should be the same assessors as the first evaluation.
We thank the reviewer for the insightful comments. We agree that the validation criteria and standards for the study should be elaborated. We have now specified this in the method section line 169
We further agree that the level of experience of the three raters along with information on which rater performed the intra-rater evaluations should be stated in the paper. We have now added the level of experience for all raters in the methods section starting at line 169. We have also specified which raters performed the intra-rater evaluation as suggested at line 169.
Second, the authors should mention their definition of extremity lymphedema instead of generically say in the inclusion criteria, “Unilateral arm lymphedema diagnosed by a lymphedema physiotherapist”. It is not clear which protocol of CDT the patients followed before the ICG lymphography (Compression sleeve, compression garment, night compression, pneumatic compression?) and if all patients evaluated were compliant with the CDT protocol.
We agree that the CDT protocol should be elaborated. We have now specified the CDT protocol prior to inclusion line 72.
The conservative treatment compliance (use of compression sleeve, compression gauntlet, night compression and pneumatic compression device) are specified in table 2, line 233.
Third, please add information about when the tape measurements were taken and by who. The authors should clarify which statistical analysis they used for the correlation between the two ICG lymphography staging and the lymphedema size. It is interesting to note in Figure 5C that in the MDA scale, the correlation with the lymphedema size is consistent from stage 0 to stage 4 and then drop in stage 5; the authors should discuss this point.
We agree that information about the timing of tape measurements (in relation to the other measurements) and who performed the tape measurements should be detailed to aid the reader in interpreting the validity of the findings. We have now added this information to line 161. For clarity, we have now also added this information to the ICG-L section line 143 and 148.
This study is limited as measuring the lymphedema by a circumferential tape measure can significantly variability amongst measurements. A more consistent reliable mechanism to measure upper extremity lymphedema is utilizing the upper extremity's total volume via water replacement or CT volume.
We agree that many methods for assessing lymphedema excists and should be discussed in relation to our choosen modality. We have now discussed circumferential tape measurements in relation to other measurements as suggested. We have refrained from discussing CT, as it is only rarely used to assess lymphedema in the modern age. We have discussed this in the discussion section line 415.
Fourth, the authors should limit their assertions: “ICG-L is a superior tool compared to lymphoscintigraphy”, as there was no lymphoscintigraphy data compared.
We agree with the reviwer that the sentence was misleading. We were referring to the literature comparing ICG-L and lymphoscintigraphy and our subjective interpretation of this. We have now rewritten the sentence in the discussion section line 456 and hope it is more clear.
Lastly, the authors state that the two ICG lymphography staging provides unique disease information unobtainable by clinical assessment, with an MDA scale providing better disease stratification than the ADB scale. However, this statement was not justified by the data reported. The authors should provide data on how they used the information from ICG lymphography and the disease stratification from the two ICG lymphography staging in terms of patient selection for microsurgical treatment and outcome.
We agree that it would add perspective to discuss how we use ICG-L stage to stratify patient selection for microsurgical treatments. We have now added this to the discussion section line 437.

Reviewer 2 Report
Overall
Extensive syntax and grammar editing should be accomplished. I suggest a careful revision of the entire manuscript to improve readability.
Abstract
The purpose of the study should be clearly stated in this section
Line 30 “size of lymphedema”. I would suggest to rephrase this. Something like “volume of the affected extremity” or “size of the affected limb” would be more appropriate. Please revise it throughout the entire manuscript
Line 23 “Three authors performed interrater, intrarater, and interscale assessments using the 23 MD Anderson (MDA) and the Arm Dermal Backflow (ADB) grading systems…...”
“Assessment” of what? This should be clarified in this section
Introduction
Line 45-46. “The Arm Dermal Backflow scale (ADB)[5,6], and the MD. Anderson classification (MDA)[1,2] are two ICG-L staging systems can both be used for ICG lymphangiography.” The meaning of this sentence is clumsy, so it should be revised.
Line 54 “Therefore, we sought to validate both staging systems in a large cohort of BCRL patients and hypothesized that the ICG-L stage would correlate well with the clinical disease severity.”
Please clarify the meaning of this sentence and revise it
Patients and Methods
Table 2 is presenting data in a quite confusing manner. Please revise it
Line 23 “Three authors performed interrater, intrarater, and interscale assessments using the MD Anderson (MDA) and the Arm Dermal Backflow (ADB) grading systems for a total of 2607 assessments.”
This information should be provided in Patients methods section of the manuscript, not only in abstract.
Line 83-93. The words in brackets should be deleted.
Discussion
Line 338. “In our study we found that lymphedema size only slightly correlated with ICG-L stage”. Do you think that other methods of arm volume assessment (i.e. Water displacement, three-dimensional
laser scanner) would result is a stronger correlation? Please discuss this point through the followings
- de Sire A, et al. Three-dimensional laser scanning as a reliable and reproducible diagnostic tool in breast cancer related lymphedema rehabilitation: a proof-of-principle study. Eur Rev Med Pharmacol Sci. 2020 Apr;24(8):4476-4485.
-Karges JR, et al. Concurrent validity of upper-extremity volume estimates: comparison of calculated volume derived from girth measurements and water displacement volume. Phys Ther. 2003 Feb;83(2):134-45.
Line 341 “patients in this study had received lymphedema treatment by a physiotherapist prior to inclusion in the study”. Is this an inclusion criterion? I could not find it in Patients and methods section
Author Response
Reviwer 2:
Overall
Extensive syntax and grammar editing should be accomplished. I suggest a careful revision of the entire manuscript to improve readability.
We have now carefully revised the readability of the manuscript. We hope that you will find our changes satisfactory.
Abstract
The purpose of the study should be clearly stated in this section
We have now clearly stated the purpose of this study in line 20 and line 25.
Line 30 “size of lymphedema”. I would suggest to rephrase this. Something like “volume of the affected extremity” or “size of the affected limb” would be more appropriate. Please revise it throughout the entire manuscript
We agree that the size of lymphedema should be rephrased for clarity. We have now rephrased “size of lymphedema” and its derivative phrases to “volume of lymphedema” throughout the manuscript. The reviwer suggest to use wording such as “volume of the affected extremity” or “size of the affected limb”, however we believe this can be misinterpreted as the volume of the entire limb. The definition of the lymphedema volume in this study is the volume of the affected arm (example 2500mL) minus the volume of the unaffected limb (example 2300mL). In this example, the lymphedema volume measures 200mL. We hope this change will be acceptable.
Line 23 “Three authors performed interrater, intrarater, and interscale assessments using the 23 MD Anderson (MDA) and the Arm Dermal Backflow (ADB) grading systems…...”
“Assessment” of what? This should be clarified in this section
We agree that this sentence was inaccurate. We have now rewritten the abstract line 25.
Introduction
Line 45-46. “The Arm Dermal Backflow scale (ADB)[5,6], and the MD. Anderson classification (MDA)[1,2] are two ICG-L staging systems can both be used for ICG lymphangiography.” The meaning of this sentence is clumsy, so it should be revised.
We agree that this sentence was clumsy. We have now rewritten the sentence for clarity in line 47.
Line 54 “Therefore, we sought to validate both staging systems in a large cohort of BCRL patients and hypothesized that the ICG-L stage would correlate well with the clinical disease severity.”
Please clarify the meaning of this sentence and revise it
We agree that this sentence was clumsy and inaccurate. We have now rewritten the sentence to clarify the meaning in line 58.
Patients and Methods
Table 2 is presenting data in a quite confusing manner. Please revise it
We have now sought to clarify the data in table 2 by providing explanatory variables and subheadings. We hope these changes are acceptable.
Line 23 “Three authors performed interrater, intrarater, and interscale assessments using the MD Anderson (MDA) and the Arm Dermal Backflow (ADB) grading systems for a total of 2607 assessments.”
This information should be provided in Patients methods section of the manuscript, not only in abstract.
We agree and have now provided this information in the method section as well in line 189.
Line 83-93. The words in brackets should be deleted.
Words in brackets have now been removed from the “In addition, we registered the following demographic information for each patient:…” paragraph in the method section line 97.
Discussion
Line 338. “In our study we found that lymphedema size only slightly correlated with ICG-L stage”. Do you think that other methods of arm volume assessment (i.e. Water displacement, three-dimensional
laser scanner) would result is a stronger correlation? Please discuss this point through the followings
- de Sire A, et al. Three-dimensional laser scanning as a reliable and reproducible diagnostic tool in breast cancer related lymphedema rehabilitation: a proof-of-principle study. Eur Rev Med Pharmacol Sci. 2020 Apr;24(8):4476-4485.
-Karges JR, et al. Concurrent validity of upper-extremity volume estimates: comparison of calculated volume derived from girth measurements and water displacement volume. Phys Ther. 2003 Feb;83(2):134-45.
Thank you for your insightfull comment. We have now discussed the agreement between tape measurements and other measurements in the discussion section line 415
Line 341 “patients in this study had received lymphedema treatment by a physiotherapist prior to inclusion in the study”. Is this an inclusion criterion? I could not find it in Patients and methods section
Thank you for your attentive comment. We agree that this information was missing in the inclusion criteria section. This have now been added in line 72.

Reviewer 3 Report
The inter-rater agreement is high and having three reviewers/raters adds objectivity to the image analysis described.
The imaging used is able to look at valocity but not at whether or not the lymph is pumping. The study is also limited by only three statis time points (0, 10 and 60 minutes after dye injection) rather than continuously viewing. In a sense the investigators can only look at the static site where dye is located at these three time points, which may not reflect the entirety of lymphedematous states throughout the limb.
One of the key findings is that lymphedema size/swelling may not be as reliable as ICG-L data for staging and directing treatment for lymphedema.
I do not know why the investigators to not use or even discuss the ISL lymphedema classifications or correlate these--other publications have incorporated ISL grade I-IV classifications and how/if they correlate with the MDA/ADB classifications and I think this would add to the paper.
A major limitation of the study is that there was no baseline arm measurement for the affected arm, adn so the lymphedematous volume of the affected arm was determined only as a delta from the volume of the healthy arm. As these subjects are established lymphedema patients, it is possible that the contralateral/unaffected arm volume may not be the most reliable comparison for hte affected arm. There have been multiple publications regardin gthis (Aldrich et al, 2012, Biomet Opt Express; Bains et al, 2015, BJS; Burnand 2012, Clin Nucl Med; Pain 2004, J Nucl Med; Pain 2004, Eur J Surg oncol). While the investigators cannot undo this limitation, there should be a significant discussion.
Another major issue is that all of these patient had lymphedema for more than one year (in fact the median duration of lymphedema was 4.5 years. As such, it is uncertain to me how/if this is relevant for new onset lymphedema and diagnosing lymphedema and its general relevance is uncertain.
Author Response
Reviwer 3:
The inter-rater agreement is high and having three reviewers/raters adds objectivity to the image analysis described.
The imaging used is able to look at valocity but not at whether or not the lymph is pumping. The study is also limited by only three statis time points (0, 10 and 60 minutes after dye injection) rather than continuously viewing. In a sense the investigators can only look at the static site where dye is located at these three time points, which may not reflect the entirety of lymphedematous states throughout the limb.
We thank the reviewer for the comments and for reviewing this manuscript. We agree with these limitations of the grading system and we agree that a continuously viewing of the lymph flow up to 60min would likely show lymph pumping mechanisms. While this was beyond the scope of the study to investigate, we have now discussed this in the discussion 365
One of the key findings is that lymphedema size/swelling may not be as reliable as ICG-L data for staging and directing treatment for lymphedema.
I do not know why the investigators to not use or even discuss the ISL lymphedema classifications or correlate these--other publications have incorporated ISL grade I-IV classifications and how/if they correlate with the MDA/ADB classifications and I think this would add to the paper.
We did stage all patients using the ISL grading at assessment. However we originally omitted this information from the paper to focus on the validity of the ICG-L classification. We do agree with the reviwer that an ISL grading and comparison to MDA/ADB classification adds to the paper. We have now added this information to the method section line 168, results section line 305 and discussion section line 437.
A major limitation of the study is that there was no baseline arm measurement for the affected arm, adn so the lymphedematous volume of the affected arm was determined only as a delta from the volume of the healthy arm. As these subjects are established lymphedema patients, it is possible that the contralateral/unaffected arm volume may not be the most reliable comparison for hte affected arm. There have been multiple publications regardin gthis (Aldrich et al, 2012, Biomet Opt Express; Bains et al, 2015, BJS; Burnand 2012, Clin Nucl Med; Pain 2004, J Nucl Med; Pain 2004, Eur J Surg oncol). While the investigators cannot undo this limitation, there should be a significant discussion.
We agree that it would be beneficial to have baseline arm measurements. However this limitation is inherent to all BCRL literature and baseline arm measurements at the time of breast cancer treatment is highly unfeasible for routine clinical use and would hamper the translational value of the study. We appreciate the suggested publications regarding predisposition of BCRL and have now discussed this in line 401 including the suggested references.
Another major issue is that all of these patient had lymphedema for more than one year (in fact the median duration of lymphedema was 4.5 years. As such, it is uncertain to me how/if this is relevant for new onset lymphedema and diagnosing lymphedema and its general relevance is uncertain.
We agree that is is a consideration for readers. However we choose not to include patients during their first year of BCRL diagnosis, because CDT and compressions garments are fitted during this time, which can affect BCRL arm volume. Patients most often seek surgical treatment for BCRL after they are conservatively optimized, making this study highly clinical relevant. In fact the duration of BCRL is broadly represented in this study following the first year. We have now specified the distribution of BCRL durations in table 2. We have now also discussed this in the discussion section line 401

Round 2
Reviewer 2 Report
The authors responded to my previous observations; they have made a significant effort to improve their manuscript. The structure has been improved. However, the language is still far from optimal.
The following issues should be addressed
Abstract
“There was an unsatisfactory agreement between the volume of lymphedema and ICG-L stage by MDA (spearmans rho = 0.44, p<0.001) and ADB (rs = 0.35, p<0.001) stage.”
Please revise the structure of this sentence to improve clarity.
Materials and methods
Line 72 “The arm lymphedema after completed decongestive therapy by a lymphedema physiotherapist.”
This sentence should be revised. There is no verb
Line 200 “Standard derivation” is a typing mistake
Author Response
Reviwer 2:
The authors responded to my previous observations; they have made a significant effort to improve their manuscript. The structure has been improved. However, the language is still far from optimal.
The following issues should be addressed
Abstract
“There was an unsatisfactory agreement between the volume of lymphedema and ICG-L stage by MDA (spearmans rho = 0.44, p<0.001) and ADB (rs = 0.35, p<0.001) stage.”
Please revise the structure of this sentence to improve clarity.
We have now carefully revised the readability of the manuscript and improved the language in several paragraphs. We hope that you will find our changes satisfactory.
We agree that this sentence was clumsy in the abstract. We have now rewritten the sentence in the abstract line 32 and hope the sentence is more clear.
Materials and methods
Line 72 “The arm lymphedema after completed decongestive therapy by a lymphedema physiotherapist.”
This sentence should be revised. There is no verb
We agree there was a verb missing in line 72. We have now corrected the sentence in line 72 with the missing verb
Line 200 “Standard derivation” is a typing mistake
We agree there was a spelling mistake in line 200. We have now corrected the spelling of standard deviation in line 200.
